# Emerging Roles of Using Small Extracellular Vesicles as an Anti-Cancer Drug

**DOI:** 10.3390/ijms241814063

**Published:** 2023-09-14

**Authors:** Hyeon Su Joo, Ju Hun Suh, Chan Mi So, Hye Jin Jeon, Sol Hee Yoon, Jung Min Lee

**Affiliations:** School of Life Science, Handong Global University, 558 Handong-ro, Buk-gu, Pohang 37554, Republic of Korea

**Keywords:** extracellular vesicle, exosome, cancer therapy, drug delivery system, cancer biomarker, tumor microenvironment

## Abstract

Small extracellular vesicles (sEVs) are emerging as a novel therapeutic strategy for cancer therapy. Tumor-cell-derived sEVs contain biomolecules that can be utilized for cancer diagnosis. sEVs can directly exert tumor-killing effects or modulate the tumor microenvironment, leading to anti-cancer effects. In this review, the application of sEVs as a diagnostic tool, drug delivery system, and active pharmaceutical ingredient for cancer therapy will be highlighted. The therapeutic efficacies of sEVs will be compared to conventional immune checkpoint inhibitors. Additionally, this review will provide strategies for sEV engineering to enhance the therapeutic efficacies of sEVs. As a bench-to-bedside application, we will discuss approaches to encourage good-manufacturing-practice-compliant industrial-scale manufacturing and purification of sEVs.

## 1. Introduction

Small extracellular vesicles (sEVs), as nanoscale double-layered particles, originate from cells, typically ranging in size between 30 and 150 nm [1]. The contents and the composition of the membrane vary depending on the origin of the sEVs. Biomolecules including proteins, lipids, and various nucleic acids are contents of sEVs, surrounded by a membrane which includes lipid rafts, integrins, and tetraspanin proteins [2]. When sEVs are absorbed by recipient cells, the cells exhibit various changes such as signaling pathway and transcription alterations [3]. The formation of sEVs is through the inward budding of multivesicular endosomes (MVEs). Thereafter, sEVs are released into the extracellular space through fusion between the MVEs’ membrane and cell surface [4]. sEVs are present in almost every biological fluid of the body, such as cerebrospinal fluid, blood, saliva, and urine [5]. In many scientific articles, sEVs and exosomes are interchangeably used. According to the guidelines of the International Society for Extracellular Vesicles, authors are suggested to use operational terms for EV subtypes by size such as small EVs (<100 nm or <200 nm) and medium/large EVs (>200 nm) instead of using exosomes [6]. sEVs have gained widespread attention in the field of cancer research owing to the cellular modulatory effects of sEVs. In patients, sEVs can contain disease-specific biomolecules. Because of the biological characteristics of sEVs, these can be used as a non-invasive diagnostic tool [7].

Although immune checkpoint inhibitors such as PD-1 blockade have shown great clinical outcomes, the response rate remains relatively low in cold tumors [8]. Also, current cancer treatments such as chemotherapy and radiation therapy can cause oxidative stress, inflammatory responses, and DNA damage, which results in side effects including cardiotoxicity, neurotoxicity, and mucositis [9].

As a natural carrier, sEVs possess better efficacy and bioavailability, lower immunogenicity, and lesser toxicity compared to other biological carriers such as lipid nanoparticles. Furthermore, sEVs can penetrate the blood–brain barrier [5]. By delivering DNA/RNA, proteins, small molecules, and other compounds, sEVs are being used as drug delivery platforms for treating cancer and other diseases.

In the context of cancer progression, sEVs have the ability to carry oncogenic molecules that stimulate cell growth, immune evasion, and blood-vessel formation, facilitating tumor progression [10]. Initially, sEVs were studied in cancer to examine their role in the change of tumor microenvironment (TME), tumor progression, and metastasis [11]. Currently, sEVs are being explored for development as an anti-cancer drug owing to their immunomodulatory activities [3,12]. Studies on sEVs in cancer are providing important aspects regarding identifying potential biomarkers, developing sEVs as drug delivery vehicles, and generating enhanced therapeutic sEVs (Table 1).

## 2. sEVs as Cancer Biomarkers

In cancer treatment, early diagnosis is crucial. Presently, tissue biopsy is the mainstay for cancer diagnosis [17]. However, tissue biopsy provides restricted information for cancer detection and monitoring due to limited sampling [18]. Liquid biopsy, in contrast to tissue biopsy, can offer non-invasive and repeated sampling for analysis. sEVs can be isolated from a wide range of body fluids, such as blood, bronchial fluid, urine, and mucus [19]. Moreover, sEVs possess a unique molecular signature that allows the identification of their parent cells. sEVs from cancer can transport specific information and cancer signaling molecules such as RNAs, DNAs, proteins, and lipids [18]. Thus, sEVs are a promising non-invasive biomarker for early cancer diagnosis [19]. Nucleic acids derived from sEVs as cancer biomarkers were primarily highlighted because there is a technical limitation for the detection of lipids and other biomolecules within sEVs [20]. The proteins and lipids derived from sEVs will be discussed briefly.

MicroRNAs (miRNAs) of cancer-derived sEVs have shown promise as biomarkers for early cancer detection in various types of cancer. Plasma samples were analyzed from colorectal cancer (CRC) patients. Four specific miRNAs (Let-7b-3p, miR-150-3p, miR-145-3p, and miR-139-3p) were identified in the plasma of cancer patients, suggesting their potential role as biomarkers for CRC diagnosis [21]. In early lung cancer detection, miRNA profiles were analyzed using plasma-derived sEVs from healthy controls, patients with small cell lung cancer (SCLC), and patients with non-small cell lung cancer (NSCLC) [22]. miRNA-483-3p showed potential as an early diagnostic biomarker for SCLC, while miRNA-152-3p and miRNA-1277-5p were identified as early diagnostic biomarkers for NSCLC [22]. Nasopharyngeal carcinoma (NPC) is usually diagnosed at a late stage and consequently shows a poor prognosis after treatment [23]. miRNA patterns in plasma-derived sEVs from patients with NPC showed that miRNA-134-5p, miRNA-205-5p, and miRNA-409-3p can be promising diagnostic biomarkers, enabling early detection of NPC [24]. In prostate cancer diagnosis, although prostate-specific antigen (PSA) has been used as a traditional biomarker, PSA is not very relevant to prostate cancer status. To test sEVs for prostate cancer diagnosis, miRNA from urinary sEVs of patients with prostate cancer were examined. An increased miR-21, miR-451, and miR-636 expression showed a significant correlation with prostate cancer [25]. Overall, sEV-derived miRNAs hold great promise as non-invasive and effective tools for early detection of cancer and prognosis assessment in various cancer types. To establish their clinical significance and application, further studies are warranted.

Many reports have focused on miRNAs derived from sEVs for cancer diagnosis. Ongoing research on the lipids and proteins derived from sEVs is underway to identify biomarkers. One study compared the lipid profile of sEVs that were obtained from individuals participating in a lung cancer screening program. The participants consisted of patients with cancer detected through screening, with benign lung nodules, or without any pathological alterations [17]. Several lipid species demonstrated distinct levels between the compared groups, with ceramide Cer (42:1) being notably upregulated in sEVs derived from cancer patients [17]. The second most common pediatric bone cancer is the Ewing sarcoma family of tumors (ESFT), and the cancer is distinguished by distinctive chromosomal EWS-ETS translocation [26,27]. To discover a new biomarker, proteomic analysis of sEVs derived from ESFT was performed. A total of 619 proteins were selected from the proteomic profile of sEVs derived from ESFT [26]. Based on the analysis, two membrane-bound proteins, CD99/MIC2 and NGFR, were selected as potential biomarkers [26]. The studies suggest that sEVs can be competitive biomarkers for the diagnosis of different types of cancers [19].

## 3. sEVs as a Drug Delivery System in Cancer

sEVs are widely used to deliver therapeutic agents, such as small molecules, miRNA, siRNAs, and proteins, owing to their biological functions and characteristics.

### 3.1. Chemotherapeutic Compound

In the past few decades, various chemotherapeutic drugs have been developed and are employed to treat a diversity of cancers. However, the main drawbacks of chemotherapy are poor bioavailability, adverse side effects, nonspecific targeting, and the development of drug resistance [28]. Numerous studies have shown that loading drugs such as doxorubicin (Dox), taxanes, lapatinib, porphyrins, and celastrol in sEVs can improve the therapeutic effects of chemotherapeutics, increasing drug accumulation in tumor tissues, and reducing systemic toxicity and side effects [29,30,31,32,33,34].

Dox has been used to treat hematologic cancers and carcinomas of the breast, lung, thyroid, and bone. However, its severe cardiotoxicity, poor pharmacokinetics, and nonspecific tissue targeting restrict its clinical applications [35]. It was reported that biodistribution changes, an increase in pharmacokinetics, and a decrease in toxicity occurred secondary to the encapsulation of Dox into liposomes [36]. Nonetheless, encapsulation-related adverse effects remain such that the circulation of liposomes in the blood may lead to adverse side effects in patients, including neutropenic fever, anemia, and skin toxicity. Several studies elucidated that improvement in vitro in the cellular toxicity of Dox encapsulated with sEVs was observed, showing an excellent cytocompatibility in comparison with free Dox [37,38,39]. Additionally, compared to the free Dox treatment, Dox encapsulated with sEVs effectively inhibited the growth of several cancer cells such as breast cancer cells, hepatoma, osteosarcoma, and glioblastoma, while significantly attenuating cardiotoxicity [37,40,41,42].

Taxanes are used to treat many solid tumors. Taxanes are poorly water soluble and are often dissolved in ethanol and Cremophor EL to improve solubility [43]. However, extreme caution should be exercised when using the solutions as they can induce hematologic toxicities including hyperlipidemia and red-blood-cell aggregation, including peripheral neuropathy [44]. One study revealed that when paclitaxel (PTX), the representative drug of taxanes, was loaded to M1 macrophage-derived sEVs, a high amount of PTX was delivered to tumor sites, activating the NF-kB pathway [45], which significantly enhanced the anti-tumor effects. Agrawal et al. showed that sEVs loaded with PTX repressed ovarian cancer cell proliferation, while PTX did not significantly inhibit ovarian cancer cell growth at the same dose [46]. These studies indicate that the anti-tumor effects of PTX loaded in sEVs can be more effective than the drug itself.

Curcumin has an anti-tumor effect, affecting apoptosis without inducing severe adverse effects, and inhibits cell proliferation, migration, and stemness [47]. Nonetheless, it has not yet been used as an anti-cancer drug owing to its poor solubility and stability in water and low bioavailability. When curcumin was loaded in milk-derived sEVs, curcumin in sEVs showed higher stability and was able to sustain harsh digestive processes [48]. Curcumin-loaded sEVs from intestinal epithelial cells showed enhanced antiproliferative activity against breast cancer, lung cancer, and cervical cancer cells compared to curcumin alone [49].

### 3.2. RNAs

RNA interference (RNAi), also known as gene silencing, has several advantages in cancer treatment [50]. However, the delivery of RNAs remains challenging, because of the RNA accumulation in the liver, nonspecific delivery, and immunogenicity of carrier vehicles. As natural drug delivery vehicles, sEVs were proposed to deliver RNAs to target tissues or cells in vivo, escaping from phagocytosis, lowering immunogenicity, and increasing efficiency (Table 2).

As shown in Table 2, sEVs can transfer RNA molecules to control cancer progression, inhibiting cancer cell proliferation, invasion, and metastasis. Although several challenges remain, including target specificity and RNA-loading efficiency, sEVs loaded with RNAs are becoming feasible in cancer therapy.

## 4. sEVs as an Active Pharmaceutical Ingredient for Cancer Therapy

sEVs, secreted by all types of cells, can be absorbed by tumor cells. When sEVs are fused with tumor cells, sEV contents exert various effects on recipient tumor cells. Depending on the type of parent cells, sEVs have different effects on tumors. For example, tumor-cell-derived sEVs have tumor-promotive effects via immune cell inhibition, stimulating tumor metastasis. sEVs from HCC contain circCCAR1. When circCCAR1 is delivered to CD8^+^ T-cells, PD-1 expression is stabilized inactivating the T-cells, leading to cancer cell survival [59]. Ovarian-cancer-cell-derived sEVs promoted metastasis through the inhibition of KLF6 resulting in the upregulation of *PTTG1* through miR-106a-5p [60]. While tumor-derived sEVs were skewed to accelerate tumor progression, sEVs from other types of cells displayed anti-tumor effects. In pancreatic cancer, miRNA let-7b-5p of natural killer (NK) cell-derived sEVs inhibited CDK6, a cell cycle regulator, hindering cell proliferation [61]. In another study, miR-3607-3p of NK cell-derived sEVs directly targeted IL-26 and inhibited the malignant transformation of pancreatic cancer [62]. In neuroblastoma, it was revealed that proteins within NK-cell-derived sEVs, such as perforin, granzyme A, granzyme B, and granulysin functioned in combination to induce cytotoxicity to CHLA255, a neuroblastoma cell line [63]. Mesenchymal stem cells (MSCs) are a promising source of sEVs due to their immunomodulatory effects. miR-598 delivered by bone marrow MSC (BM-MSC)-derived sEVs targeted mRNA of *THBS2*. Through blocking THBS2 transcription, NSCLC migration and proliferation were inhibited [64]. In acute myeloid leukemia (AML), BM-MSC-derived miR-222-3p within sEVs induced the downregulation of IRF2/INPP4B signaling and increased apoptosis of the AML cell line THP-1 [65]. Peroxiredoxins are antioxidant proteins and known to be related to tumor metastasis and inflammation in CRC [66]. MiR-431-5p carried by umbilical cord MSC-derived sEVs demonstrated inhibitory effects on *PRDX1*, including restricted CRC cell line growth [67]. LINC00622 included in adipose-derived stem cell-derived sEVs promoted gamma-aminobutyric acid B-type receptor 1 and inhibited neuroblastoma cell growth [68]. Plant-derived sEVs also demonstrated anti-tumor effects. sEVs from tea flowers induced reactive oxygen species (ROS) generation, resulting in cell-cycle arrest. Therefore, proliferation and migration of breast cancer cells were halted [69]. Similar to the tea flower, lemon-derived sEVs showed anti-tumor effects on gastric cancer cells through ROS generation [70]. Cannabidiol-containing sEVs from *Cannabis sativa* induced mitochondrial-dependent apoptosis of two HCC cell lines—HepG2 and Huh-7 [71]. The data suggest that plant-derived sEVs can be effective and be potential candidates for cancer therapy.

The TME consists of immune cells, surrounding blood vessels, fibroblasts, and extracellular matrix [72]. The TME hinders anti-cancer drug accessibility through the provision of a physical barrier and inhibits immune cell functions with inhibitory signaling molecules. With these features, the TME reduces the therapeutic effects of anti-cancer drugs for solid tumors [73]. Therefore, TME regulation is indispensable for a successful clinical outcome. sEVs are being tested to modulate the TME because sEVs can access the TME easily and carry biomolecules between TME cells, which can be used to treat solid tumors surrounded by the TME [74]. Tumor-associated macrophages (TAMs) are major immune cells in the TME and exert immunosuppressive effects. sEVs from M1 macrophages switched M2-polarized TAMs to M1-type macrophages, potentiating anti-tumor activity [75]. NK cell-derived sEVs impacted other immune cells. miR-10b-5p, miR-92a-3p, and miR-155-5p of NK cell-derived sEVs induced Th1 polarization of CD4^+^ T-cells. When T-cells absorbed sEVs, *GATA3* mRNA was downregulated, *TBX21* was activated, and thereby T-cells were polarized to IFN-γ and IL-2 producing Th-1 cells [76]. miR-182 of MSC-derived sEVs stimulated immune responses in a ccRCC mouse model and promoted maturation of T-cells in vitro [77]. Ginseng-derived sEVs stimulated CCL5 and CXCL9 secretion from TAMs and recruited CD8^+^ T-cells to TME, reducing tumor size [78].

## 5. Therapeutic Potential of sEVs with Immune Checkpoint Inhibitors (ICIs)

The combination of chemotherapy and ICIs has been administered to synergistically enhance anti-tumor immune responses and improve survival in certain patients [79]. Several studies comparing sEVs with ICIs indicated that sEVs can serve as an anti-cancer drug modality [80,81,82,83,84,85,86]. In 2019, Hong et al. targeted high-molecular-weight hyaluronan (HA) in the TME with sEVs carrying GPI-anchored PH20 hyaluronidase (Exo-PH20) [80]. Exo-PH20 effectively degraded HA into low-weight HA and activated CD103^+^ dendritic cells (DCs), leading to the activation of tumor-specific CD8^+^ T-cells. A single injection of Exo-PH20 suppressed tumor growth more effectively than a single injection of an anti-PD-L1 antibody. A combination treatment of Exo-PH20 with anti-PD-L1 antibody showed synergistic anti-cancer effects in a melanoma mouse model [80]. Matsuda et al. loaded siRNA into milk-derived sEVs to target β-catenin signaling. A single injection of siRNA-loaded sEVs suppressed tumor growth and induced T-cell infiltration in an HCC mouse model [81]. Treatment of sEVs with anti-PD-1 antibody demonstrated synergistic anti-tumor effects [81]. Another study suggested xenogenization of cancer cells through sEVs expressing fusogenic viral antigens [82]. Engineered HEK293T-derived sEVs (mVSVG-Exo) stimulated DCs through a TLR4-dependent manner, leading to the activation of CD8^+^ T-cells and CD4^+^ T-cells of tumor-draining lymph nodes in a lymphoma mouse model. Although the anti-PD-L1 antibody has no effects on lymphoma, mVSVG-Exo dramatically inhibited tumor progression [82]. To stimulate DCs, Park et al. employed synthetic bacterial sEVs (SyBV) as a cancer vaccine [83]. DCs engulfed SyBV and were activated through an increased cytokine secretion and co-stimulatory molecule expressions. In a melanoma mouse model, a single injection of SyBV suppressed tumor growth more compared with an anti-PD-1 antibody treatment. The anti-tumor effects were enhanced when SyBV was combined with an anti-PD-1 antibody [83]. Chen et al. engineered sEVs to express high-affinity variant human PD-1 protein (havPD-1 EVs) and suggested that havPD-1 EVs themselves could act like ICIs [84]. The affinity to PD-L1 and anti-tumor effects of havPD-1 EVs were compared with humanized anti-PD-L1 antibody. In a human breast cancer mouse model, havPD-1 EVs showed similar anti-tumor effects to an anti-PD-L1 antibody. Additionally, by loading a cancer chemotherapy drug (senaparib), researchers enhanced the therapeutic efficacy of havPD-1 EVs [84].

A recent study targeted highly expressing CD38 through CD38 siRNA-loaded sEVs (EVs/CD38) [85]. While a single treatment of an anti-PD-L1 antibody showed no anti-tumor effects, sEVs loaded with CD38-targeting siRNA suppressed tumor growth and induced M1 macrophage polarization in tumor tissue of an HCC mouse model [85]. Another research team loaded antigens (a-galactosylceramide, and ovalbumin) into bone marrow DC-derived sEVs to activate NK cells and NK T-cells [86]. A single treatment of antigen-loaded sEVs (A-sEVs) sufficiently decreased tumor volume and increased survival rate in a melanoma mouse model. Furthermore, pre-treatment with A-sEVs induced memory T-cell responses, resulting in an improved survival rate. When compared to the anti-cancer effects of A-sEVs, a single treatment of anti-PD-1 or anti-PD-L1 antibodies revealed no effects in the same mouse model. Combined treatment of A-sEVs with ICIs suggested that anti-cancer effects are derived mainly from A-sEVs [86].

## 6. Engineering sEVs for Enhanced and Targeted Cancer Therapy

Modifications of sEVs can be performed to either parent cells, isolated sEVs, or both. Engineering of parent cells can enhance the specificity of sEV tumor targeting [84,87,88]. Recently, Wang et al. endeavored to enhance the delivery efficiency of therapeutic agents to hepatocarcinoma which was enriched in GPC3 [87]. They genetically modified HEK293 cells to double knock out endogenous *GPC3* and *B2M* by CRISPR/Cas9. Subsequently, through viral transfection, anti-GPC3 single-chain variable fragments (scFv) and fusogens were co-expressed on the cell surface. Finally, each therapeutic agent (PTX, gelonin, and siR-Sox2) was loaded to nanovesicles. The modified nanovesicles (eFT-CNVs) exhibited improved efficiency for delivery compared to the administration of each therapeutic agent in a free form. Notably, among eFT-CNVs, eFT-CNVs loaded with PTX demonstrated enhanced effectiveness in the in vivo HepG2 xenograft model reducing the tumor volume by 84.1-fold compared to free PTX [87]. It is known that increased levels of HA in the TME increase interstitial pressure and hinder drug delivery [80]. Feng et al. transfected HEK293T-cells with a recombinant plasmid encoding PH20 (human hyaluronidase) followed by sEV isolation [88]. Subsequently, the sEV exterior was modified by coating sEVs with 1,2-distearoyl-sn-glycero3-phosphoethanolamine-N-[folate(polyethylene glycol)], resulting in the surface modification of sEVs with DSPE-PEG incorporated into the lipid layer, and FA was displayed (Exos-PH20-FA). Exos-PH20-FA degraded HMW-HA into low-molecular-weight (LMW)-HA and this LMW-HA induced M1 polarization of macrophages. Exos-PH20-FA also reduced tumor-cell metastasis through its tumor-targeting effects. In a 4T1-cell-bearing mouse breast-cancer model, both local and systemic injection of Exos-PH20-FA remarkably reduced tumor size. Moreover, lung metastasis was more effectively suppressed compared to the group treated with Exos-PH20 without FA [88]. The utilization of sEVs as carriers for delivering therapeutic nucleic acids holds potential for cancer treatment [33,89,90,91,92,93]. The introduction of nucleic acids into parent cells can allow the generation of modified sEVs containing the introduced nucleic acids. The modification process can be performed without compromising the sEV membrane integrity [94]. Liu et al. transduced cells with lentivirus-miR-7-5p to produce miR-7-5p containing sEVs (exo-miR-7-5p) [90]. In a human lung cancer mouse model, the combined exo-miR-7-5p and everolimus administration resulted in a significant tumor volume reduction when compared to the administration of exo-miR-7-5p or everolimus alone. MiR-7-5p displayed synergistic therapeutic efficacy with everolimus in NSCLC by inducing apoptosis and inhibiting the MMK/eIF4E axis and mTOR pathway, leading to anti-cancer effects [90]. As the hepatocyte growth factor (HGF) has been known to stimulate both cancer cell and vascular cell growth, the HGF-cMET pathway emerged as a viable target for clinical interventions [95]. Zhang et al. transfected HGF siRNA into HEK293T-cells [91]. In the SGC7901 xenograft mice model, the tumor size and mass were significantly decreased when sEVs containing HGF-targeting siRNA were administered. Additionally, the tumor angiogenesis process was suppressed [91].

Once sEVs are isolated from parent cells, sEVs can be subjected to surface engineering using direct approaches [32,88,96,97,98,99,100,101,102]. Nie et al. coated sEVs derived from M1 macrophages with CD47 and SIRPa antibodies [96]. To this end, M1 macrophage-derived sEVs and CD47/SIRPa antibodies were subjected to incubation with gentle rotation overnight (M1 Exo-Ab). The coated antibodies were designed to be cleaved and subsequently released in the TME due to acidic conditions. In the 4T1 tumor-bearing BALB/c mice model, the M1 Exo-Ab revealed tumor volume reduction. During the observation period, all mice that received M1 Exo-Ab survived, and no lung metastasis was detected [96]. Based on the understanding that A33 is a highly prevalent antigen found in cancer, Master et al. obtained sEVs from A33-positive LIM1215 cells (A33-Exo) and incorporated Dox into the sEVs [97]. They coated surface-carboxyl superparamagnetic iron oxide nanoparticles with A33 antibodies to facilitate the specific binding of A33 antibodies on the sEV surface to A33-positive Dox-loaded sEVs. This complex was designed to specifically target A33-positive colon cancer cells [97]. The use of cancer-cell-derived sEVs as a cancer vaccine holds promise in eliciting immune responses. Because sEVs from cancer cells have tumor antigens, identifying or purifying specific tumor antigens may not be feasible [103]. Furthermore, introducing an adjuvant to sEVs can efficiently stimulate anti-tumor responses [104]. According to Morishita et al., immunization with tumor-cell-derived sEVs carrying tumor antigens (endogenous antigens) and an adjuvant (CpG DNA) demonstrated anti-tumor effects in tumor-bearing mice [98]. To load CpG DNA on the sEV surface, streptavidin (SAV) encoding plasmid DNA was transfected to B16BL6 cells, and sEVs were isolated (SAV-exo). SAV-exo was then incubated with biotinylated CpG DNA to generate CpG DNA-loaded sEVs (CpG-SAV-exo). CpG-SAV-exo enhanced the antigen presentation of DC2.4 cells, leading to Th-1 antigen-specific immune responses [98]. Loading of therapeutic agents directly into sEVs is a beneficial approach involving a relatively simple modification process [46,105,106,107,108,109]. Agrawal et al. reported that orally delivered PTX-loaded milk-derived sEVs (ExoPAC) improved anti-cancer effects and reduced toxicity related to PTX [46]. sEVs were isolated from pasture-raised Jersey cow-derived milk, and PTX was loaded into sEVs simply by mixing the PTX solution. ExoPAC was stable in the gastrointestinal fluids, and the toxicity test showed no significant toxicity. Furthermore, in the xenograft athymic nude mice model bearing human lung cancer cell lines A546, the ExoPAC oral gavage group resulted in a 29% or greater reduction in tumor growth compared to an equal-sized free PTX-treated group [46]. Group O red blood cells (RBCs) can serve as a universal donor. Because group O RBCs are easily accessible in blood banks and lack DNA [110], group O RBCs can be a suitable source for the large-scale generation of sEVs [105,111]. Usman et al. introduced oncogenic miR-125b targeting antisense oligonucleotide (ASO) into group O RBCs through electroporation. The modified sEVs showed that the sEVs were successful in inhibiting miR-125b in MOLM13, an AML cell line [105]. In the AML xenograft mouse model, the delivery of miR-125b ASO through O-RBC-sEVs effectively reduced the development of leukemia which was confirmed by reduced bioluminescent signals of leukemia. Cas9 mRNA and mir-125b-2 gRNA loaded in O-RBC-sEVs resulted in a 98% reduction in miR-125b in the recipient MOLM13 cells 48 h after treatment [105].

## 7. Large-Scale Expansion of Cell Sources for Anti-Cancer sEV Production

### 7.1. Large-Scale Cell Culture System

For industrial-scale sEV manufacturing, the development of a culture system for the parent cells is indispensable. Recent studies on large-scale cell culture manufacturing for sEV mainly focus on culturing MSCs [112,113,114,115,116,117,118,119,120,121]. Studies suggested various types of three-dimensional (3D) bioreactor for culturing human cells efficiently with a high-density suspended form [122].

Several studies attached MSCs to microcarriers and then inoculated and cultured them in a stirred-tank [113], spinner-flask [116], or vertical-wheel bioreactor [115,118] with agitation at a low rpm. Other studies used a hollow-fiber bioreactor [112,114] or rocking bioreactor [117,121] for MSC expansion. Results showed a significantly increased production of sEVs from MSCs through a 3D culture in a bioreactor compared to a two-dimensional (2D) culture [113,114,115,116,117,118,121]. Through an in vitro functional assay, 3D-cultured MSC-sEVs showed increased angiogenic [113], neurogenic [115], and anti-inflammatory [121] potentials. Bellio et al. showed that 3D-cultured MSC-sEVs improved cardiac function in a myocardial infarction mouse model [112]. Although a bioreactor was not used, Kim et al. [119] and Lim et al. [120] used specialized plates to form an MSC spheroid. Studies suggest that a 3D bioreactor could be applied for industrial-scale production of anti-cancer sEVs from various cell types. Cells which are 3D-cultured show changed cellular physiologies compared to 2D-cultured cells. In MSCs, a 3D culture increased stem cell gene expression (Oct4, Sox2 and Nanog) [121]. MSC-derived sEV (MSC-sEVs) production also increased through a 3D culture, with expression of mRNA related to sEV biogenesis (such as CD9, CD62, CD81, Alix, TSG, HRS, syntenin, and Rab27a) upregulation [121]. A 3D culture changed protein and miRNA cargo composition in MSC-sEVs [121]. Another study reported that a 3D-culture induced MSC growth and MSC-sEV production which enhanced neuronal growth properties [115]. The osteogenic potency of MSC-sEVs is enhanced through a 3D culture [123]. The studies implicate that a 3D culture could enhance therapeutic efficacy and yield anti-cancer sEVs. Moreover, researchers should be alerted that the cell phenotypes can be altered through a 3D culture compared to a traditional 2D culture.

### 7.2. Scalable Purification of sEVs from Massive Cell-Cultured Media

Several comparative studies suggested that purification methods affect yield, population, and the therapeutic efficacy of sEVs [124,125,126,127,128]. A position paper from the International Society for Extracellular Vesicles categorized the purification methods of sEVs in terms of point of recovery and specificity [6]. Using precipitation with polymers (such as PEG or others), low-molecular-weight cutoff (MWCO) centrifugal filters and high-speed ultracentrifugation (UC) are expected to harvest sEVs with a high rate of recovery, but low specificity. Size-exclusion chromatography (SEC), high MWCO centrifugal filters, differential UC, tangential flow filtration (TFF), and membrane-affinity column methods are positioned on a moderate rate of recovery and specificity. Depending on the purpose, methods can be selected.

In studies using a 3D bioreactor, MSC-sEVs were purified through differential UC [112,114,115], density-gradient UC with polyethylene glycol-6000 [116,117,118,121], or a combination of SEC with TFF [113,129] from large-scale cell culture media. In a laboratory scale, several studies reported various sEV purification methods such as asymmetric depth-filtration [130], microfluidic SEC [131], flow field-flow fractionation [132,133], elasto-inertial flow focusing [134], immunoaffinity chromatography [135], heparin affinity chromatography [136,137], and anion exchange chromatography [138]. For clinical application, a GMP-compliant method for appropriate sEV purification should be established to obtain sEVs with high yield and high purity.

## 8. Conclusions

sEVs are emerging as promising agents for cancer immunotherapy. As an avatar of parent cells, cancer-derived sEVs hold great potential as cancer biomarkers. As natural carriers, sEVs are efficient and bioavailable, have low immunogenicity, and are less toxic as a drug delivery system. sEVs can deliver chemotherapeutic compounds and anti-cancer RNAs to the targeted cancer cell. Moreover, sEVs themselves can act as an active pharmaceutical ingredient (API), which indicates that proteins, lipids, and nucleic cargos in sEVs can exert anti-tumor effects. Immune cells or MSC-derived sEVs show potential in suppressing tumor growth directly or in remodeling the TME from an immunosuppressive (cold tumor) to an immune-stimulated condition (hot tumor) through immune cell activation. sEV engineering strategies using surface modification or API loading could enhance the therapeutic potential of sEVs. Single or combinatorial [139] treatment of sEVs with other cancer therapies, especially ICIs, could be expected as a promising anti-cancer strategy. The establishment of GMP-compliant industrial-scale cell culture, purification, and characterization methods for the production of anti-cancer sEVs is necessary for clinical application.

## Figures and Tables

**Table 1 ijms-24-14063-t001:** Ongoing clinical trials in the recent 5 years.

Study ID	Study Title	Study Status	Conditions	Interventions	Start Date	Locations	Publications
NCT05375604	A Study of exoASO-STAT6 (CDK-004) in Patients with Advanced Hepatocellular Carcinoma (HCC) and Patients with Liver Metastases from Either Primary Gastric Cancer or Colorectal Cancer (CRC)	Terminated, Phase 1	Advanced HCC|Gastric Cancer Metastatic to Liver|Colorectal Cancer Metastatic to Liver	Drug: CDK-004	June 2022	City of Hope National Medical Center, US	Not provided
NCT03608631	iExosomes in Treating Participants with Metastatic Pancreas Cancer with KrasG12D Mutation	Recruiting, Phase 1	KRAS NP_004976.2:p.G12D|Metastatic Pancreatic Adenocarcinoma|Pancreatic Ductal Adenocarcinoma|Stage IV Pancreatic Cancer AJCC v8	Drug: mesenchymal stromal cell-derived exosomes with KRAS G12D siRNA	January 2021	M.D. Anderson Cancer Center, US	[13]
NCT05559177	An Open, Dose-escalation Clinical Study of Chimeric Exosomal Tumor Vaccines for Recurrent or Metastatic Bladder Cancer	Recruiting, Early Phase	Recurrent or Metastatic Bladder Cancer	Biological: chimeric exosomal tumor vaccines	September 2022	Shanghai Pudong Hospital, China	Not provided
NCT04499794	The Study of Exosome EML4-ALK Fusion in Non-Small Cell Lung Cancer (NSCLC) Clinical Diagnosis and Dynamic Monitoring	Recruiting	Untreated Advanced NSCLC Patients|FISH Identified ALK Fusion Positive or Negative	Drug: ALK inhibitor	August 2020	Cancer Institute and Hospital, Chinese Academy of Medical Sciences, China	[14]
NCT05563766	A Phase II Trial to Evaluate the Effect of Itraconazole on Pathologic Complete Response Rates in Resectable Esophageal Cancer	Recruiting, Phase 2	Esophageal Adenocarcinoma|Esophageal Squamous Cell Carcinoma|Gastroesophageal Junction Carcinoma	Drug: itraconazole	July 2023	VA Palo Alto Health Care System, US	Not provided
NCT05705583	A Companion Diagnostic Study to Develop Circulating Exosomes as Predictive Biomarkers for the Response to Immunotherapy in Renal Cell Carcinoma	Recruiting	Renal Cell Carcinoma	Other: blood and urine collection	January 2023	Zhejiang Cancer Hospital, China	[15]
NCT05575622	Clinical Study for Combined Analysis of CTC and Exosomes on Predicting the Efficacy of Immunotherapy in Patients with HCC	Recruiting	HCC	DsEVICE: CTC PD-L1, exosomal PD-L1, and exosomal LAG-3 detection	November 2023	Zhongnan Hospital, China	Not provided
NCT05625529	ExoLuminate Study for Early Detection of Pancreatic Cancer	Recruiting	Pancreas Cancer|Exosomes|Extracellular Vesicles|Pancreatic Neoplasms		December 2022	Biological Dynamics, US	[16]
NCT05854030	Serum Exosomal microRNA (miRNA) Predicting the Therapeutic Efficiency in Lung Squamous Carcinoma	Recruiting	Lung Neoplasm|Squamous Cell Carcinoma|Exosomes	Diagnostic test: collect plasma samples and clinical features	April 2022	Tianjin Medical University Cancer Institute and Hospital, China	Not provided
NCT04529915	Multicenter Clinical Research for Early Diagnosis of Lung Cancer Using Blood Plasma-Derived Exosome	Enrolling by invitation	Lung Cancer	Diagnostic test: exosome sampling	April 2020	Korea University Guro Hospital, South Korea	Not provided
NCT04357717	ExoDx Prostate Evaluation in Prior Negative Prostate Biopsy Setting	Terminated, Phase 1	Prostate Cancer	Diagnostic test: ExoDx Prostate	March 2020	Chesapeake Urology Research Associates, Baltimore, Maryland	Not provided
NCT03811600	Exosomes Implication in PD1-PD-L1 Activation in OSAS	Completed	Sleep Apnea Syndromes, Obstructive|Cancer	Diagnostic test: PD1/PD-L1 exosomal expression	March 2019	Laboratoire du Sommeil, France	Not provided
NCT04499794	The Study of Exosome EML4-ALK Fusion in NSCLC Clinical Diagnosis and Dynamic Monitoring	Recruiting	Untreated Advanced NSCLC Patients|FISH Identified ALK Fusion Positive or Negative	Drug: ALK inhibitor	August 2020	Cancer Institute and Hospital, Chinese Academy of Medical Sciences, China	Not provided
NCT04629079	Improving the Early Detection of Lung Cancer by Combining Exosomal Analysis of Hypoxia with Standard of Care Imaging	Recruiting	Lung Cancer		October 2020	Borthwick Research Unit, Lister Hospital, Stevenage, SG1 4AB, United Kingdom	Not provided
NCT05101655	Construction of Microfluidic Exosome Chip for Diagnosis of Lung Metastasis of Osteosarcoma	Completed	Osteosarcoma|Pulmonary Metastases		October 2020	Ruijin Hospital Shanghai Jiao Tong University School of Medicine, China	Not provided

**Table 2 ijms-24-14063-t002:** Anti-cancer effects of sEV-based RNAi-mediated gene silencing.

RNA Molecule	Targeting Cancer/Cell Type	Source of sEVs	Mechanism	References
miR-155	Oral cancer cells	Oral squamous cell carcinoma	Directly targeting *FOXO3a*, reducing cisplatin resistance	[51]
miR-144	Non-small cell lung cancer (NSCLC)	Bone-marrow-derived MSC	Downregulation of *cyclin E1* and *cyclin E2*, resulting inhibition of cell proliferation, colony formation, and number of S phase-arrested cells	[52]
miR-199a-3p	Ovarian cell lines (CaOV3, SKOV3, OVCAR3), xenograft OC mouse model	Primary omental fibroblasts of OC patients	Upregulation of miR-199a-3p expression level	[53]
siRNA-tLyp-1	HeLa, HEK293T, NSCLC	HEK293T	Downregulation of *tLyp-1*	[54]
miRNA-29c-3p	Orthotopic xenograft mouse model	Primary omental fibroblasts of OC patients	Upregulation of MMP2, weakening OC metastasis	[55]
miR-497	A549, human umbilical vein endothelial cell (HUVEC)	HEK293T	Targeting *YAP1*, *HDGF*, *CCNE1*, *VEGF-a*, inhibited peritoneal dissemination in OC mice model, and diminished cMet expression, ERK phosphorylation, and MMP2 expression	[56]
siRNA or etomoxir	Colon cancer	Cell culture medium	Downregulation of CPT1A, reversed sensitivity, increased tumor apoptosis, and decreased proliferation of tumor cells	[57]
circ-0051443	HCC	Human liver cell line (HL-7702)	Protection of BAK1 and inhibition of the malignant behavior of HCC by sponging miR-331-3p	[58]

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
