# Peer review of "Emerging Roles of Using Small Extracellular Vesicles as an Anti-Cancer Drug"

_ijms, 2023, doi:10.3390/ijms241814063_

Round 1

Reviewer 1 Report

This review is a very interesting summary of the topic regarding the importance of small extracellular vehicles (sEVs) as a novel therapeutic strategy for cancer therapy. Based on the current knowledge the authors discussed how sEVs can exert tumor-killing effects or modulate the tumor environment leading to anticancer response. They give many examples of sEVs as diagnostic tools, drug delivery systems, and active pharmaceuticals. The most important information was well summarized in tables. The work is well-thought, well-organized, and includes most published articles. Generally, I have not found significant limitations in this manuscript; conversely, I think that it has many strengths, such as an interesting theme and an accurate presentation of the issue.

I have only a few minor remarks:

In the Introduction section, I would add a few sentences about the problems of cancer treatment, which would better justify the topic.

Table 1: I think there should be any reference(s) here as for Table 2.

Line 289: Explain ‘API’ at the first use.

There are some editing errors like lack of spaces that should be corrected. In the References section, I can see a lack of the journal’s title in many places (see references 3, 5, 4, etc.)

English is nice, minor corrections are suggested.

Author Response

We appreciate the editor and the reviewer for their positive comments regarding our manuscript. We also thank them for their helpful and constructive suggestions. We have incorporated changes to the manuscript that address the reviewer’s comments, as outlined below.

  1. In the Introduction section, I would add a few sentences about the problems of cancer treatment, which would better justify the topic.

We thank for reviewer’s suggestion. We add sentences to explain the limitation of current cancer therapy for convincing the importance of out topic at Line 36-39.

  1. Table 1: I think there should be any reference(s) here as for Table 2.

We appreciate the reviewer for suggesting the important point. We add references that clinical study was cited.

  1. Line 289: Explain ‘API’ at the first use.

We apologize with our mistake. We add explanation of API (active pharmaceutical ingredient) at Line 321.

  1. There are some editing errors like lack of spaces that should be corrected. In the References section, I can see a lack of the journal’s title in many places (see references 3, 5, 4, etc.)

We appreciate for the pointing critical mistake. We check and fix editing errors. And we also correct lack of journal’s title in Reference section.

Reviewer 2 Report

The draft presented by Hyeon Su Joo and collaborators on the use of small extracellular vesicles as anticancer drugs is an interesting, well-organized and very clear review. The authors describe various applications of sEVs, both as biomarkers of cancer and as drug delivery systems and active tools for therapy. Recent strategies for engineering sEVs to improve cancer therapy are also reported. Interestingly, approaches for the production and purification of sEVs are also discussed.

Therefore, the review is of potential interest to readers of the journal.

To improve the quality of the presentation I suggest some minor changes:

More details on the physicochemical properties of sEVs (size, heterogeneity, composition, etc.) should be given in the introductory section.

Also, the difference between sEVs and other commonly used delivery devices, such as exosomes, should be clearly explained.

In Table 1, is it possible to refer to the literature for the studies cited?

Also, in the last column, the status of the institution that conducted the study should be specified.

Finally, I suggest including 1-2 figures to schematize the processes described (however, this is not mandatory).

Some typos to correct

Author Response

We appreciate the editor and the reviewer for their positive comments regarding our manuscript. We also thank them for their helpful and constructive suggestions. We have incorporated changes to the manuscript that address the reviewer’s comments, as outlined below.
  1. More details on the physicochemical properties of sEVs (size, heterogeneity, composition, etc.) should be given in the introductory section. Also, the difference between sEVs and other commonly used delivery devices, such as exosomes, should be clearly explained.

We thank for reviewer’s comment for better context. We add sentences to describe more detail properties of sEVs and the difference between sEVs and exosomes in definition at Line 23-33.

  1. In Table 1, is it possible to refer to the literature for the studies cited? Also, in the last column, the status of the institution that conducted the study should be specified.

We thank for reviewer’s suggestion. As reviewer’s comment, we add references that the clinical study was cited. And we specify the status of the institution by adding country which study was performed.
